# The Association of Self-Reported ADHD Symptoms and Sleep in Daily Life of a General Population Sample of School Children: An Inter- and Intraindividual Perspective

**DOI:** 10.3390/brainsci12040440

**Published:** 2022-03-25

**Authors:** Lilly Buhr, Tomasz Moschko, Anne Eppinger Ruiz de Zarate, Ulrike Schwarz, Jan Kühnhausen, Caterina Gawrilow

**Affiliations:** 1Workgroup School Psychology, Department of Psychology, Faculty of Science, University of Tübingen, 72076 Tübingen, Germany; tomasz.moschko@uni-tuebingen.de (T.M.); anne.eppinger@uni-tuebingen.de (A.E.R.d.Z.); ulrike.schwarz@uni-tuebingen.de (U.S.); caterina.gawrilow@psycho.uni-tuebingen.de (C.G.); 2LEAD Graduate School & Research Network, University of Tübingen, 72076 Tübingen, Germany; jan.kuehnhausen@med.uni-tuebingen.de; 3IDeA—Interdisciplinary Research Center for Individual Development and Adaptive Education, Goethe University Frankfurt, 60323 Frankfurt am Main, Germany; 4Department of Child and Adolescence Psychiatry, Psychosomatics and Psychotherapy, University Hospital of Tübingen, 72076 Tübingen, Germany

**Keywords:** attention-deficit/hyperactivity disorder, sleep, ambulatory assessment, multilevel analysis

## Abstract

Sleep and Attention-Deficit/Hyperactivity Disorder (ADHD) have repeatedly been found to be associated with each other. However, the ecological validity of daily life studies to examine the effect of sleep on ADHD symptoms is rarely made use of. In an ambulatory assessment study with measurement burst design, consisting of three bursts (each 6 months apart) of 18 days each, 70 German schoolchildren aged 10–12 years reported on their sleep quality each morning and on their subjective ADHD symptom levels as well as their sleepiness three times a day. It was hypothesized that nightly sleep quality is negatively associated with ADHD symptoms on the inter- as well as the intraindividual level. Thus, we expected children who sleep better to report higher attention and self-regulation. Additionally, sleepiness during the day was hypothesized to be positively associated with ADHD symptoms on both levels, meaning that when children are sleepier, they experience more ADHD symptoms. No association of sleep quality and ADHD symptoms between or within participants was found in multilevel analyses; also, no connection was found between ADHD symptoms and daytime sleepiness on the interindividual level. Unexpectedly, a negative association was found on the intraindividual level for ADHD symptoms and daytime sleepiness, indicating that in moments when children are sleepier during the day, they experience less ADHD symptoms. Explorative analyses showed differential links of nightly sleep quality and daytime sleepiness, with the core symptoms of inattention and hyperactivity/impulsivity, respectively. Therefore, future analyses should take the factor structure of ADHD symptoms into account.

## 1. Introduction

People with attention deficit/hyperactivity disorder (ADHD) have self-regulation difficulties and frequently experience symptoms of inattention, hyperactivity and impulsivity [1,2]. These problems concern around 3.4% of children worldwide [3]. With an ADHD diagnosis in childhood the probability of negative life outcomes concerning health, vocational, and social areas increases [4]. High self-regulation on the other hand is associated positively with academic achievement, healthy behaviors and interpersonal relationships [5]. Therefore, it seems necessary to understand antecedents and correlates of ADHD, to tailor practices for therapy or prevention and enhance self-regulation. Today, theory assumes that ADHD is caused by a combination of biological, psychological, and social factors [2]. However, the question how daily life circumstances affect individuals’ ability for attention and regulation of behaviour has only scarcely been researched. The study at hand therefore investigates how ADHD symptoms in a general population sample of German schoolchildren are associated with sleep. Thereby, variance cannot only be found with help of clinical samples but also in a general population sample, since people differ in their ability to self-regulate and most people experience at least some ADHD symptoms from time to time.

### 1.1. Dimensionality of ADHD Symptoms

Current theories define self-regulation as dimensional in nature. According to this dimensional view, every person lies on a continuum between two extreme poles of high ADHD symptoms on the one side and high self-regulation in behaviour on the other side [6,7]. Taking that into account, children should not be categorised into those with an ADHD diagnosis and those without the disorder, but differ on the extent of their capability to self-regulate attention and behaviour [8]. Supporting that view, research has found that ADHD symptoms in the general population depict a normal distribution ranging from high attention and self-regulation of behaviour to extreme inattention, hyperactivity and impulsivity as well for children as for adults [9,10]. To depict the whole continuum of attention and self-regulation of behaviour, research should therefore consider differences in and correlates of ADHD symptomatology in a general population sample instead of applying group comparisons.

### 1.2. Fluctuations in ADHD Symptomatology

Recent findings suggest that differences in self-regulation do not only exist between individuals (interindividual; between-person) but ADHD symptoms also fluctuate within individuals (intraindividual; within-person) [11]. Ambulatory assessment studies are the golden standard to capture these moments of high and low symptomatology [12]. Thereby, participants indicate their current experiences repeatedly, for example several times per day on a digital device, like smartphones or tablets [13]. Self-regulation capacities and ADHD symptoms fluctuate highly in the daily lives of children with and without ADHD diagnosis over days and weeks [14,15]. These fluctuations as well as their preceding and following events need to be investigated more thoroughly to better understand the disorder. Indicating which events and experiences lead to better self-regulation of cognition (i.e., attention) and behaviour (i.e., impulsivity) could help to improve the daily lives of people with high levels of ADHD symptoms [16]. One phenomenon which has been shown to be associated to cognitive and behavioural measures like executive functions and therefore might also be related to ADHD symptoms is sleep [17].

### 1.3. Importance of Sleep

Sleep is an important factor for cognitive and psychological functioning in daily life [18]. Sleep is defined as a state with highly diminished consciousness and responsiveness, while brain activity can still be high [19]. It is assumed that this brain activity is crucial for memory construction as well as restoration of body and brain tissue. Lack of sleep might thus impair emotion regulation and cognitive functions [20]. Therefore, it has often been hypothesized that sleep might also impact the capability to self-regulate ones’ behaviour and thereby influence symptoms of inattention, hyperactivity, and impulsivity e.g., [21].

### 1.4. The Relation between Sleep and ADHD Symptomatology

According to the state regulation model, children with sleep loss might not have the energy to adequately regulate their arousal and activation [22]. A few studies have implemented sleep restriction and extension experiments, where children followed a strict sleep schedule including significantly less or more sleep than their average sleeping hours. When seven- to eleven-year-old children slept one hour shorter than usual, their teachers described them as less emotionally stable and more hyperactive/impulsive. In the opposite condition, when children slept one hour longer than normally, they were rated as more alert and showed more emotional stability [23]. In another study implementing a similar intervention of sleep restriction, this intervention functioned as a moderator of response inhibition and self-regulation in preschool children. With normal sleep schedules, children who showed higher response inhibition applied more self-regulation strategies while playing with an unsolvable puzzle. After the sleep restriction, no association between response inhibition and self-regulation strategies was found [24]. Given this empirical evidence, consequently longer or better sleep should have positive consequences on childrens’s self-regulation. When parents of five to twelve year old children with an ADHD diagnosis received a behavioral sleep intervention, which consisted of psychoeducation concerning sleep hygiene practices and standardised behavioural strategies, ADHD symptom levels of the children six months later showed a significantly greater decrease than those of a control group [25]. These findings indicate that there might be an effect of sleep on the ability for attention and self-regulation of behaviour on the between-person level. Thus, children sleeping more and better than others might experience less ADHD symptoms. However, while restricting or extending individuals’ sleep under laboratory conditions mirrors typically occurring, intraindividual fluctuations of sleep quality in daily life, to our knowledge no study has explicitly investigated the intraindividual associations between sleep and ADHD symptoms in daily life up to now. Intraindividual fluctuations describe the changes which happen within an individual, for example a child might sleep very good in one night and experience bad sleep in the next. This has to be distinguished from interindividual differences, the between-person difference, where one child has in general better sleep than the other. Both, inter- and intraindividual differences should be considered when investigating the effect of sleep on ADHD symptoms.

Besides the quality of night sleep, which might influence the regulation of attention and behaviour, there also might exist an effect of the current personal experience of tiredness during a specific moment of the day. Although this might seem paradoxical, children who are feeling sleepy could be more instead of less active than usual, as it indeed has been described by many parents [26]. The feeling of sleepiness might therefore lead to more hyperactive and impulsive symptoms. This observation can also be explained with support of the state regulation model: the evaluation mechanism of the individual might register a state of underarousal due to sleepiness, and therefore react with an enhanced hyperactivity/impulsivity [22]. The state of tiredness might also interfere with attention, since children do not possess the energy to regulate their cognition and behaviour adequately. Thus, it is important to examine the daily life of individuals to disentangle how natural fluctuations in sleep quality and tiredness during the day interact with the fluctuations of ADHD symptoms.

### 1.5. Measurement of Sleep

The overarching construct of sleep seems to be composed of several different sleep indicators like sleep duration, sleep efficacy, or sleep quality. These indicators in turn might be measured by calculating the hours of total sleep time, the number of awakenings, the time needed to fall asleep (sleep onset latency), and the subjective feeling of being rested in the morning [27]. All of these indicators might thereby be related to other aspects of human functioning. Past research has for example found groups of children with and without ADHD diagnosis to differ in sleep onset latency (the time needed to fall asleep) and sleep efficiency but not in the number of awakenings during the night or the actual hours of being asleep [28]. Sleep onset latency has been found to be related to night awakenings, deeper sleep, subjective sleep quality and longer sleep [29]. This would make sleep onset latency an economical and short indicator of sleep quality in general.

Another aspect of sleep is the feeling of being tired or sleepy during the day. According to a meta-analysis, this daytime sleepiness has shown higher correlations to school performance than sleep quality and sleep duration [30]. This indicates that feelings of sleepiness might be partly independent from the actual sleeping time but still have an impact on self-regulation [26]. Consequently, researchers should be aware of these different parameters when deciding for an index to measure sleep.

### 1.6. The Current Study

Considering the above-described research, we were interested in examining how self-reported nightly sleep quality, sleepiness over the day, and ADHD symptoms interact with each other on a between- as well as a within-person level in the daily life of German schoolchildren. ADHD symptomatology was defined on a dimensional level, therefore a general population sample was gathered to depict as much variance in the construct as possible. To account for fluctuations in the measured constructs, ambulatory assessment was used. Both constructs were examined through self-report. Sleep quality was defined by a combination of sleep onset latency and subjective sleep quality. Daytime sleepiness was assessed through indication of the activation level. In the current study, 10–12-year-old children were asked to report on their sleep, sleepiness, and ADHD symptoms on 18 consecutive days. These assessment periods were repeated three times, each time half a year apart, resulting in a maximum of 54 days of assessment. Such an ambulatory assessment not only decreases memory bias, but also ensures a high ecological validity [12], and allows to determine both, interindividual differences between the children, as well as intraindividual fluctuations over time.

Building on the state regulation theory as well as on previous findings about the relationship of sleep and ADHD symptoms, we expected the following effects: we predicted a negative relationship between self-rated night sleep quality and self-rated ADHD symptoms on (1) the between-person level across all assessments, and (2) on the within-person level (relation between prior night sleep quality and following day ADHD symptoms). ADHD symptoms should be higher for children who on average sleep worse than other children (interindividual difference), and be higher after a night of worse sleep than a child usually has (intraindividual fluctuation). Further, we expected a positive relationship between self-rated daytime sleepiness and self-rated ADHD symptoms (3) on a between-person level across all assessments, as well as (4) on a within-person level. Children who are sleepier in general are supposed to experience more ADHD symptoms (interindividual difference), and in moments when a child is more tired than usual it is expected to indicate more symptoms (intraindividual fluctuation).

## 2. Method

Data was collected within the research project “Adaptive dynamics of cognitive and behavioral variability in children with symptoms of attention deficit /hyperactivity disorder (AttentionGO!)”, an intensive longitudinal study which was conducted at the Department of School Psychology at the University of Tübingen in cooperation with the Goethe University, Frankfurt. The project was funded by the German Research Foundation (project number GA 1277/9-1) and approved by the ethics committee of the German Society for Psychology (DGPs, CG 102018_amd_112013). The Ministry of Culture, Youth, and Sport in Baden-Württemberg, Germany, approved recruitment in schools (file number 31-6499.20/1087). The present study refers to three measurement bursts (each lasting 18 days), which took place between autumn 2017 and autumn 2018.

### 2.1. Participants

Participants were recruited in seven schools in southern Germany (*n* secondary school = 5, *n* community school = 2). The sample consisted of a total of 70 pupils in grade 5 (55.71 % female). The age range of the children at the beginning of the study period was 10 to 12 years (M = 10; 9 years, SD = 5.7 months). Eight of the participating children had a diagnosis of ADHD, all of them were receiving medical treatment. Exclusion criteria consisted of psychological health (no other diagnosed psychological disorder than ADHD). Figure 1 shows the recruitment process and retention of the participants throughout the study period. Parents and children learned about the possibility to participate in the project through presentations at their respective schools and registered via school. Participation was voluntary and only possible with the written consent of the children and their parents. Participants could end their participation in the study at any time without giving reasons. As compensation for their participation, each family received a voucher worth 40€ for an excursion of their choice (e.g., swimming pool, zoo).

In order to prevent any conclusions regarding personal data, all collected data was pseudonymised and stored in a password-protected manner on internal servers of the University of Tübingen. The participating persons were informed about the type of data storage, the handing over of data on request and the deletion of data in accordance with the Basic Data Protection Regulation (DSGVO).

#### Procedure

For the length of each 18-day survey period, the participating children were given a smartphone (Motorola MotoG4plus©). Children were trained in school to use the smartphones and fill out the daily questionnaires. It was made sure that all items and instructions were understood by the participants. Each survey period started on a Wednesday and we used a time-contingent sampling method. Smartphones rang three times a day (i.e., in the morning directly after getting up, in the afternoon after school, and in the evening before going to bed) within specific time ranges which were adapted according to individual schedules of the participants. Assessment times could vary on weekends to better fit into the lives of the children. After the signal, children had up to 30 min to participate, otherwise the occasion was indicated as missing. Children were asked to give information about their current ADHD symptoms as well as their current feeling of sleepiness on all three assessment moments per day and to indicate their sleep quality during the prior night in the morning measurement directly after getting up.

The study protocol of the ambulatory assessment phase was similar for all three bursts, which were administered approximately half a year apart. However, within the project an intervention to enhance self-regulation was conducted before Burst 2. Children were assigned to one of two groups, with the experimental group receiving the full intervention and the control group receiving a reduced intervention. Both groups showed slight improvement in their self-regulation with no significant difference between the groups [14].

### 2.2. Measures

The ambulatory assessment design poses specific challenges to the scales which are used within scientific studies. First, their wording has to be in such a way that repetitive assessment actually captures fluctuations in the concepts. Therefore, in the current projects all instructions included the phrase “Since the last time I filled in the form…”. Second, participant burden is already very high due to a long study period. Consequently, scales have to be as short as possible to minimize disruption of the daily life of participants and keep compliance rates as high as possible. To account for these concerns, all scales used in the present study were tested in a pilot study. Only items which proved to depict substantial variance were included in the study. Additionally, we tried to apply broad questionnaires, to assess as many research questions as possible without excessive extension of each assessment occasion. In the following, we will describe the adapted items which are relevant for our research questions.

#### 2.2.1. ADHD Symptoms

Four items of the children self-report version of the Conners C3-AI Scales [31] on attention and behaviour were modified for daily recording (“Since the last time I filled in the form I talked too much.”; “Since the last time I filled in the form I forgot what I was supposed to do.”; “Since the last time I filled in the form I had too much energy to sit still.”; “Since the last time I filled in the form I could hardly concentrate.”). The self-report scales are suited for children of eight to 18 years of age [31]. The children indicated on a Likert scale how much the statements applied to them since the last assessment (1 = not at all to 6 = exactly). High values therefore expressed high ADHD symptom levels. To obtain an ADHD score for each measurement time point, we calculated averages across all four items for each moment the child answered at least three of the four items. We computed multilevel reliability estimates using generalizability theory analyses [32] to determine the reliability of these scores to capture individual differences (between-person reliability *R_KF_*; 0.98–0.99) (As initial sighting of the data indicated that there might be substantial differences in children’s ADHD symptom levels between bursts we computed the reliability estimates for the ADHD scale separately for each burst), as well as day-to-day fluctuations in symptom levels (within-person reliability *R_C_*; 0.61–0.69). Multilevel reliability estimates for only afternoon assessments was 0.93–0.97 (between-person), and 0.59–0.70 (within-person). Additionally, we checked for validity by comparing our modified version of the Conners scales for the ambulatory assessment with the standardized measures (without modification) of the Conners ADHD index score that the children filled out in school before each burst. Mean ADHD scores from daily assessment within each study burst are associated weakly but significantly with the child’s ADHD index score, assessed at the beginning of each measurement burst, respectively. More specifically, there was a weak correlation for Burst 1, *r*(53) = 0.47, *p* < 0.001, for Burst 2, *r*(41) = 0.31, *p* = 0.040, and for Burst 3, *r*(34) = 0.42, *p* = 0.011, indicating that higher mean ADHD scores from daily assessment within each study burst were associated with higher ADHD index scores. Therefore, we concluded that the modified ADHD scales were valid to measure the construct we intended.

#### 2.2.2. Sleep

The items for self-report of subjective sleep quality were adapted from the study by Könen and colleagues [33]. Children rated their sleep quality of the previous night on a Likert scale from one (poor) to six (good). The time taken to fall asleep was also recorded on a Likert scale from one (long) to six (not long). Thus, a high value of the duration of falling asleep indicated that children fell asleep quickly. To calculate a sleep quality score, the average of both items was computed, with higher scores indicating better night sleep quality. Between-person reliability for this score was 0.97, and within-person reliability was 0.54.

#### 2.2.3. Daytime Sleepiness

To indicate their current affect, children filled out a slightly modified and shortened version of the Multidimensional Mood Questionnaire MDMQ [34]. Eight items were answered on individual 6-point Likert scales. The daytime sleepiness was calculated by averaging the following two items: (1.) “At the moment I feel tired (1) or well rested (6)” and (2.) “At the moment I feel sleepy (1)—awake (6)” Between-person reliability for this score was 0.99, and within-person reliability was 0.81.

### 2.3. Statistical Analysis

All analyses for the current research question were preregistered (Doi:10.17605/OSF.IO/T9XEA, https://archive.org/details/osf-registrations-t9xea-v1, submission: 8 December 2021). The data was processed and analysed with help of the programme R [35] version 4.1.1., using the nlme package (version 3.1.-153) to conduct multilevel regression analyses. To analyze between- and within-person associations between children’s night sleep quality and ADHD symptom levels the following day (Hypothesis 1 and 2), we used a multilevel model including a random intercept and random slopes for time and within-person fluctuations in sleep quality e.g., [36]. Due to the specific assessment design, we used several time variables. Data is nested within bursts, which were administered each half a year apart. To account for this nested structure and possible trends in missing data, variables were included to account for the 18 days within each burst and the respective differences in results of Burst 2 and Burst 3 compared to Burst 1. We expected missing data to be higher on weekends and additionally assumed differences in sleep quality between weekends and weekdays. Therefore, we included weekend as a control variable into the models. For this specific analysis, we paired night sleep quality ratings assessed in the morning and rating of ADHD symptom levels assessed the following afternoon. To avoid biased results just due to extreme individual reports of either night sleep quality, or ADHD symptom levels on certain days, we considered data points that lie three standard deviations above or below a participant’s individual mean across time as outliers and excluded them from all data analyses. To differentiate the effects of within-person fluctuations from trait-like individual differences in sleep quality, we split the raw scores into two components: a between-person component indicating individual i’s trait-like tendency for better/worse sleep than other individuals (this between-person component was calculated by subtracting the sample’s grand mean from each person mean (a) participants average across all study days). The grand means for all variables of interest (Table 1) were obtained by calculating the average of all person means, and a within-person component indicating individual i’s tendency on day t to have slept better/worse than usual. To facilitate the interpretation of results and comparison of within- and between-person effects, we divided the predictor (within-person fluctuations and between-person differences in sleep quality) by the between-person standard deviation across the study period to identify small, moderate, and large effect sizes in standard deviation units [37]. Based on previous findings, we included gender, age and ADHD medication as control variables in the model without specific hypotheses. Equation (1) describes the full model tested:ADHD_it_ = (*γ*_00_ + *u,*_*i*0_) + (*γ*_01_ + *u*_*i,*1_*)* Time_it_ + *γ*_02_ SleepB_i_ + (*γ*_03_ + *u*_*i*2_) SleepW _it_ + *γ*_04_ Weekend_it_ + *γ*_10_ Burst2_it_ + *γ*_11_ Burst2_it_ × Time_it_ + *γ*_12_ Burst2_it_ × SleepB_i_
*+ γ*_13_Burst2_it_ × SleepW _it +_
*γ*_14_ Burst2 _it_ × Weekend_it_ + *γ*_20_ Burst3_it_ + *γ*_21_ Burst3_it_ × Time_ti_ + *γ*_22_ Burst3_it_ × SleepB_i_ + *γ*_23_ Burst3_it_ × SleepW_it_ + *γ*_24_ Burst3 _it_ × Weekend_it_ + *γ*_30_ Gender_i_ + *γ*_31_ Age_i_ + *γ*_32_ Medication_i_ + *ε_it_*(1)

Using this equation, we tested whether the following fixed effects differ from 0:

(a) an intercept, *γ*_00_, representing the average level of ADHD symptoms on study day 1 during Burst 1;

(b) an average linear time trend, *γ*_01_, indicating the change in ADHD symptom levels over the 18 study time days during Burst 1, centered on Day 1;

(c) the between-person effect of sleep quality during Burst 1, centered at the sample’s grand mean in sleep quality across all three bursts, *γ*_02_, indicating the difference in ADHD symptom levels for participants with better sleep quality of one unit (i.e., one between-person standard deviation in sleep quality), compared to the typical participant’s sleep quality;

(d) the within-person effect of sleep quality during Burst 1, centered at the participant’s personal mean in sleep quality across all three bursts, *γ*_03_, indicating the change in ADHD symptom levels on days following night with better sleep of one unit (i.e., one between-person standard deviation in sleep quality) than the participant’s usual level in sleep quality;

(e) the weekend effect, *γ*_04_, indicating the mean difference in ADHD symptom levels on weekend days (i.e., Saturday and Sunday; coded 1), and school days (i.e., Monday to Friday; coded 0);

(f) the difference in the mean level of ADHD symptoms on study day 1 in Burst 2 (coded 1) compared to Burst 1 (coded 0), *γ*_10_;

(g) the difference in the average linear time trend in Burst 2 (coded 1) compared to Burst 1 (coded 0), *γ*_11_;

(h) the difference in the between-person effect of sleep quality in Burst 2 compared to Burst 1, *γ*_12_;

(i) the difference in the within-person effect of sleep quality in Burst 2 compared to Burst 1, *γ*_13_;

(j) the difference in the weekend effect in Burst 2 compared to Burst 1, *γ*_14_;

(k) the difference in the mean level of ADHD symptom on study day 1 in Burst 3 (coded 1) compared to Burst 1 (coded 0), *γ*_20_;

(l) the difference in the average linear time trend in Burst 3 (coded 1) compared to Burst 1 (coded 0), *γ*_21_;

(m) the difference in the between-person effect of sleep quality in Burst 3 compared to Burst 1, *γ*_22_;

(n) the difference in the within-person effect of sleep quality in Burst 3 compared to Burst 1, *γ*_23_;

(o) the difference in the weekend effect in Burst 3 compared to Burst 1, *γ*_24_;

(p) the effect of children’s gender, *γ*_30_, indicating the mean difference in ADHD symptom levels between boys (coded 1), and girls (coded 0);

(q) the effect of children’s age, *γ*_31,_ indicating the difference in ADHD symptom levels for older participants of one unit (month);

(r) the effect of ADHD medication, *γ*_32_, indicating the mean difference in ADHD symptom levels between children receiving ADHD medication (coded 1), and children not receiving ADHD medication (coded 0).

The model in Equation (1) also tested whether the following between- and within-person random effects differ from 0:

(s) *u*_0*i*_ captures how much a particular participant deviates from the average intercept (i.e., random intercept);

(t) *u*_1*i*_ captures how much a particular participant deviates from the average time slope (i.e., random time slope);

(u) *u*_2*i*_ captures how much a particular participant deviates from the average within-person effect (i.e., random sleepiness slope);

(v) *ε_it_* indicates how much a particular participant’s ADHD symptom levels on a gives study time point deviates from the value predicted by their person-specific regression line (i.e., residual error).

We allowed for a maximal random effects structure with covariances of all random effects. To account for the intensive longitudinal data structure, we modeled time dependence of the residuals with a first-order autoregressive structure AR1; e.g., [36]. Model analyses were conducted with restricted maximum likelihood estimation and a probability level of *p* < 0.05 to indicate significance of effects based on t-values of each model coefficient.

Likewise, we tested the between- and within-person associations between children’s daytime sleepiness and their ADHD symptom levels (Hypothesis 3 and 4), using children’s sleepiness and ADHD symptom ratings collected three times a day—that is, on up to 54 study time points per burst—with between-person differences and within-person fluctuations in daytime sleepiness rather than sleep quality as predicting variable within the regression model.

## 3. Results

### 3.1. Descriptive Results

The number of possible observations was calculated by multiplying 18 days with 3 bursts for all 55 children that were recruited in November 2017 and with 2 bursts for the 15 children that started with the study in April 2018. This procedure ensured that those participants who were newly recruited for the second burst, were not inflating the dropout rate. Thus, there were up to 3510 observations of night sleep quality possible. As daytime sleepiness and ADHD symptom levels were assessed three times a day, this results in up to 10,530 observations, respectively. Data on night sleep quality was collected 2051 times, resulting in a participation rate of 58%, while data on daytime sleepiness was collected 5799 times (55%). In total, data on ADHD symptom levels was collected 5733 times (54%), with 1669 observations collected in the afternoon (48% of all possible observations in the afternoon).

Before further data analyses, 36 night sleep quality, 21 daytime sleepiness and 110 ADHD symptom level observations were excluded due to being defined as outliers. When only considering ADHD symptom level in the afternoon, 132 observations had to be excluded.

Within each study burst, missing values were more likely to occur on weekends compared to school days by up to 68% (Burst 1: Odds Ratio *(OR)* = 1.38, 95% CI [1.14, 1.67], Burst 2: *OR* = 1.68, 95% CI [1.40, 2.01], and Burst 3: *OR* = 1.43, 95% CI [1.17, 1.76]). Moreover, with each day within a study burst the likelihood for missing values increased by up to 6% compared to the previous day (Burst 1: *OR* = 1.06, 95% CI [1.04, 1.08], Burst 2: *OR* = 1.06, 95% CI [1.04, 1.07], and Burst 3: *OR* = 1.04, 95% CI [1.02, 1.06]).

Mean self-report of ADHD symptom levels in the sample was relatively low (*M* = 1.52, *SD* = 0.50; only afternoon: *M* = 1.62, *SD* = 0.60). Children reported medium to high sleep quality (*M* = 4.58, *SD* = 0.82). Most variance was indicated for daytime sleepiness (*M* = 2.68, *SD* = 0.98). Table 1 lists descriptive statistics of all constructs utilized for testing of hypotheses. The intraclass correlation coefficients (ICC) indicate how much of the total variance can be explained by variance on the interindividual level. Thus, around 44% of the variance in ADHD symptom levels can be explained by interindividual differences. Consequently, around 56% of the variance is composed of intraindividual fluctuations and measurement error.

Fluctuations of the variables can be inspected more thoroughly in Figure 2 for night sleep quality and in Figure 3 for daytime sleepiness. For sleepiness, the graph indicates higher values in the mornings as the evenings, as would be expected in normal circadian rhythms.

### 3.2. Multilevel Analyses

#### 3.2.1. Association of Night Sleep Quality and ADHD Symptoms

To assess the association of sleep quality during the preceding night and ADHD symptoms during the school day, multilevel models were conducted. As can be seen in Table 2, we found no significant associations between night sleep quality and ADHD symptom levels during the initial study burst, neither on the interindividual nor on the intraindividual level. This did not change during the subsequent bursts, except for a significant increase in the between-person association of night sleep quality and ADHD symptom level in Burst 2 compared to Burst 1 (*γ*_12_ = 0.18 (SE = 0.08)*, p* = 0.03), which almost annulated a non-significant trend for this between-person association in Burst 1 (*γ*_02_ = −0.16 (SE = 0.09)*, p* = 0.07). However, we found that ADHD symptoms decreased significantly during Burst 1 (*γ*_01_ = −0.27 (SE = 0.13)*, p* = 0.04), with no significant changes in within-burst decrease rates across bursts. Also, children reported significantly lower ADHD symptom level at the beginning of Burst 3 compared to Burst 1 (*γ*_01_ = −0.25 (SE = 0.11), *p* = 0.02). The random intercept showed to be significant in the analyses, indicating significant differences in children’s initial ADHD symptom levels.

#### 3.2.2. Association of Daytime Sleepiness and ADHD Symptoms

Similarly, to test for the association of self-reported daytime sleepiness and ADHD symptoms at the same time, we likewise calculated a multilevel model (Table 3). No effect of daytime sleepiness on ADHD symptoms can be seen on the interindividual level in the initial study burst. Again, we found a significant change in the between-person association of daytime sleepiness and ADHD symptom levels in Burst 2 compared to Burst 1 (*γ*_12_ = −0.12 (SE = 0.06), *p* = 0.04), which almost annulated a non-significant trend for this between-person association in Burst 1 (*γ*_02_ = −0.16 (SE = 0.10), *p* = 0.11). However, the data indicated a negative within-person association of daytime sleepiness and ADHD symptoms in the initial study burst (*γ*_02_ = −0.04 (SE = 0.02), *p* = 0.01), with no significant changes in the subsequent bursts. This result suggests that in moments when participants felt more tired during the day they indicated less ADHD symptoms, therefore contradicting our hypothesis. Also, we again found significant decreases in ADHD symptom levels throughout each study burst, as well as overall decreased symptom levels at the beginning of Burst 2, and Burst 3, compared to Burst 1, respectively. Regarding our control variables, we found increased ADHD symptom levels in children receiving ADHD medication, which overlaps with an ADHD diagnosis in our sample.

Significant effects were found for the random intercept and both random slopes, implying substantial variance in children’s initial ADHD symptom levels, variance in symptom level change within bursts, and variance in the size of the within-person association of ADHD symptom levels and daytime sleepiness.

### 3.3. Explorative Post-Hoc Analysis

After conducting all planned and pre-registered analyses, we decided to investigate children’s ADHD symptom levels separately for symptoms of inattention and symptoms of hyperactivity-impulsivity. As we will argue in the discussion section, these core dimensions of ADHD might show a unique and discriminative link to markers of sleep quality in everyday life. To this end, we conducted two separate scores for children’s levels of inattention (“Since the last time I filled in the form I forgot what I was supposed to do.”; “Since the last time I filled in the form I could hardly concentrate.”), and levels of hyperactive-impulsive behavior (“Since the last time I filled in the form I talked too much.”; “Since the last time I filled in the form I had too much energy to sit still.”), and reran our analysis in correspondence to the procedure described above, however, with separate multilevel models for levels of inattention, and hyperactive-impulsive behavior. The complete results of these post-hoc analyses are added to the appendix (Table A1 and Table A2). In summary, regarding our initial inter- and intraindividual hypotheses, we found the following results with respect to the association of night sleep quality and inattention: (a) a negative between-person association between children’s night sleep quality and levels of inattention the following day in Burst 1—that is, children who sleep better than others report to have lower levels of inattention—but the size of this association decreased significantly in Burst 2 compared to Burst 1, and in Burst 3 compared to Burst 1; and (b) a negative within-person association between children’s night sleep quality and levels of inattention the following day across all measurement bursts—that is, after sleeping better than usual, children report to have lower levels of inattention the following day. No inter- or intraindividual associations were found between night sleep quality and hyperactivity-impulsivity. Regarding the relationship between daytime sleepiness and inattention (c) a positive between-person association between children’s daytime sleepiness and levels of inattention in Burst 1—that is, children with higher levels of daytime sleepiness than others report higher levels of inattention—but the size of this association decreased significantly in Burst 2 compared to Burst 1, and in Burst 3 compared to Burst 1. There was no within-person association evident between daytime sleepiness and levels of inattention. For the core symptom of hyperactivity-impulsivity, we found (d) a negative within-person association to daytime sleepiness across all measurement bursts—that is, children report lower levels of hyperactive-impulsive behavior in moments of higher daytime sleepiness than usual. There was no between-person association evident between daytime sleepiness and levels of hyperactive-impulsive behavior.

## 4. Discussion

In the current study, we investigated the relationship of self-reported sleep variables and ADHD symptoms on a between- as well as a within-person level in German schoolchildren. With an intensive longitudinal study, applying a measurement burst design with ambulatory assessment, daily fluctuations in the constructs of sleep quality, daytime sleepiness and ADHD symptom levels were assessed. With this measurement approach we expanded the current literature on sleep and ADHD. In contrast to earlier studies, which compared groups of children with and without ADHD diagnosis, we defined ADHD symptoms on a dimensional level by using a general population sample, in line with current dimensional theories for the classification of psychological disorders [6]. Fluctuations in all constructs could be investigated further due to the repeated measurement. Finally, ecological validity was enhanced in comparison to laboratory studies by implementing ambulatory assessment in the daily life of participants with the help of smartphones.

Multilevel analyses did not confirm a relationship between sleep quality during the night and ADHD symptoms on the subsequent day on the inter- or the intraindividual level. Accordingly, we must reject our first two hypotheses, since we had expected a negative effect of sleep quality on ADHD symptoms within and between children. Although this is to our knowledge the first study to investigate the relationship of sleep and ADHD symptoms in (school) children’s daily lives, theory and previous research would have hinted to such a connection. State regulation theory implies that with worse sleep (inter- and intraindividually), children have less capacity for self-regulation and therefore show more ADHD symptoms [22]. Support for this has been found in earlier studies. For example, when their sleep was restricted for several nights within an experimental study, children experienced significantly more problems with alertness and emotional regulation [23]. In contrast to such an experimental study, where sleep restriction was externally exerted, children in our study seemed to have a relatively good night sleep and indicated an overall good sleep quality. With this lack of variance, it might have been difficult to find an effect of sleep quality on ADHD symptoms even if it was present. Additionally, earlier research findings are hinting to an effect in form of an inverted U shape of sleep on cognitive functioning, with too much sleep provoking a negative effect. After nights when children sleep either much less or much more than on average, they perform worse in a working memory task than after nights with their usual sleep length [33].

Similarly, we did not find an interindividual effect of subjective daytime sleepiness on ADHD symptoms. However, on the within-person level, we found a negative association between daytime sleepiness and ADHD symptoms. Our data implies that in moments when children feel more tired and sleepy during the day, they report less ADHD symptoms than when they feel more activated and awake. This finding is contrary to our hypothesized effect. It seems that children who are well rested also experience more energy to feel restless. One possible explication for the effect could lie in the timely structure of the study. Sleepiness and ADHD symptoms were measured three times a day. As Figure 2 shows, children indicated high sleepiness in the morning and evening but low sleepiness in the mid-day measurement. However, the time when self-regulation is most needed and therefore ADHD symptoms might be most easily detected is the time that children spend in school [14]. Therefore, it might be assumed that children were not actually able to inform about their ADHD symptoms at specific times of the day, since self-regulatory processes were not needed that strongly. In future studies, it might be interesting to examine the interaction of sleepiness and ADHD symptoms during the school day.

Especially interesting is the found effect of the three bursts on ADHD symptom levels. The findings indicate that in Burst 3 ADHD symptoms at the beginning of the burst are significantly lower than in Burst 1. In the second model, this effect can also be found for Burst 2, where children start lower than in Burst 1. The difference between the models results from the fact that in the first model only ADHD symptoms on the second moment at the middle of the day are considered. The second model uses all three indications of ADHD symptom levels on each day. Several possible explanations can be found for this effect of burst on ADHD symptom levels. The most obvious explanation might be an aging effect. Within the course of normal development, children get more attentive and learn to better self-regulate their behaviour. Therefore, symptoms of inattention and hyperactivity/impulsivity decrease with age [38,39]. Another possible cause for the effect might lie in the format of the study protocol. The study included an intervention after Burst 1, aiming at promoting self-regulatory behaviour. To this end, children were allocated to two different intervention groups (mental contrasting with implementation intentions vs. mental contrasting), with both groups showing similar improvement in self-regulation following the interventions [14]. As there were no differential intervention effects, we would not expect that the implementation of the intervention confounds the relationship between sleep quality and ADHD symptoms investigated in the current study. However, to reassure that results are not influenced by this experiment, we integrated the intervention as a control variable in a post-hoc analysis. None of the effects changed due to this additional variable as can be seen in Appendix B.

Another result we found is the significant decrease of ADHD symptoms within each of the bursts. In general, children reported significantly less symptoms at the end of the burst than in the beginning. This could be explained by an initial-elevation bias [40]. Independently of the topic of ambulatory assessment studies, self-reports are often higher in the first measurement timepoints and get more stable after a while. A support for this assumption in our data can be found in Figure 2 where higher values at the beginning of each burst are graphically depicted. It might be helpful to further investigate this effect and possibly conduct future analyses without the first few measurement timepoints.

In our original models, we integrated all three core symptoms of ADHD into one common factor to enhance the reliability of the scale. However, previous research has found that different ADHD subtypes might be associated with different sleeping patterns [17,41]. For our study this might imply that children feel less hyperactivity/impulsivity symptoms when they are tired but are at the same time more inattentive. Therefore, in an explorative post-hoc analysis, which was not preregistered, we individually examined the two ADHD symptom factors inattention and hyperactivity/impulsivity separately in models with nightly sleep quality and daytime sleepiness. We found a significant negative effect of night sleep quality on inattention on the inter- as well as on the intraindividual level. Thus, children who slept better on average indicated less inattentive symptoms in general and after a night when they slept better, children indicated less inattentive symptoms. Interestingly, the interindividual effect decreases in Burst 2 and Burst 3 respective to Burst 1. Furthermore, we found a positive interindividual effect between sleepiness and inattention; children who report in general to be more tired also report to experience more inattention. However, also this effect seems to be smaller in Burst 2 and 3 than in Burst 1. We did not find an intraindividual effect between sleepiness and inattention. For hyperactivity/impulsivity as the dependent variable, we solely found a significant negative intraindividual effect of sleepiness. Thus, in moments when children were sleepier, they indicated less hyperactive/impulsive symptoms. Other than that, no effect of sleep on hyperactive/impulsive symptoms was found.

These exploratory analyses incorporate some interesting insights into our data and the implications should be investigated more thoroughly in future research. To summarize, it seems that our hypotheses apply better for the inattentive factor of ADHD while hyperactivity/impulsivity seem not to be related to sleep and sleepiness as measured in our study, or even in the opposite direction than expected. These findings might reflect a general effect where sleep quality and sleepiness only affect attention. It might however also be an effect of the specific age group. People tend to grow calmer with age and are better able to self-regulate their behaviour with age. This effect has often been shown in ADHD research, where adults report less hyperactive/impulsive symptoms than children but still significant impairments in their attention [38]. Thus, children in our sample might already have outgrown the tendency to show more hyperactive/impulsive symptoms when they have slept badly or feel sleepier.

### 4.1. Limitations

Despite the numerous advantages the current study adds to the existing research literature, the study design also might incorporate specific drawbacks and potential for improvement.

In general, ambulatory assessment has great potential to capture daily fluctuations in ADHD symptoms and sleep of children. However, there is still a lack of adequately tested scales to use within this specific research design [32]. We tried to account for this by slightly modifying the scales and selecting only specific items which proved to show substantial variances within a pilot study. Nevertheless, future research might show that different scales are better suited to depict the fluctuations of ADHD symptoms and sleep in the daily life of schoolchildren.

Another drawback of ambulatory assessment always is the high participant burden which is put on the participants. Answering the same questions three times a day for 18 days in three different bursts is very exhausting, especially for children. Although we shortened the scales as much as possible, occasions with missing answers increased with time within each burst and many children dropped out of the study between the bursts. We tried to control for these dropouts and missing data by including burst, day within burst and weekend into the models. With compliance rates of 48–58% we received enough data to model inter- and intraindividual differences of the children. Still, future research should try to prevent this dropout effect by reducing burden and enhancing commitment of the participants.

Most obvious seems to be the question whether self-reports of sleep are a valid instrument to measure actual sleep quality in children. For measuring sleep, subjective and objective measurements all show their own advantages and drawbacks [17]. The utilization of polysomnography in sleep laboratories leads to well documented physical and neurological data but lacks ecological validity. Actigraphs can easily be worn at home in the participants’ natural environment but have their drawbacks in only measuring movement and therefore being fault-prone in indicating sleep. In sleep research with children, parent report is often used to gather information about quantity and quality of sleep, however, as children grow older and get more independent, parents’ might lose insight into their actual sleeping behaviour. Research with adults and adolescents is often relying on self-report measures of sleep. It has been found that children report more problems falling asleep and retaining sleep than their parents indicate [42]. Therefore, we were interested to see how children would self-report their quality of sleep in their daily life and how this data is related to other measures like self-reported ADHD measures [33].

Critics might object that participants of this age might lack the relevant introspection and humans in general might not be able to give valid reports of their sleep, given that the key feature of sleep is the lack of consciousness [19]. This limitation of the study should be integrated into future studies, which might use combinations of self-report with more objective measures like polysomnography or actigraphy [17]. In the current study, actigraphs were only administered throughout the day to minimize participant burden, therefore we had to rely on self-report of sleep quality during the night. This question of the amount of introspection for self-report in children of this age group might also apply the assessment of ADHD symptoms. Here as well, future research should compare these self-reports with more objective measures or parent- and teacher-rated scales to examine the validity of the children’s responses. However, since we found very high between-person reliability and high within-person reliability in our analyses, we figured the self-report scales to be adequate for the assessment of ADHD symptoms in the daily life of children.

Furthermore, we decided to measure sleep quality by combining self-reported sleep onset latency and subjective sleep quality. These constructs have shown to be related to other psychological factors in earlier studies e.g., [33] and depicted most variance in a pilot study. Given the already high participant burden in the study, questionnaires had to be as concise as possible. Other constructs indicating sleep should be investigated further. For example, the total hours of sleep could be examined [41,43]. Nevertheless, also this construct has its drawbacks, since sleep needs might differ between children. Also, number of awakenings during the night could be a good indicator of sleep. This construct however might be difficult to measure in self-report since people often do not remember their awakenings the next day.

### 4.2. Implications and Future Research

Although the current study did not confirm the hypotheses, it might bring new ideas and questions to the research area. A very positive finding is the fact that children in our general population sample indicated overall relatively high sleep quality and low ADHD symptoms throughout their daily lives. Comparing different methods to evaluate sleep quality might help to define which measurement might be related to other physiological and psychological constructs. The finding that ADHD symptoms seem to decrease over time, both between and within the bursts should be further examined. The first might be investigated in future research by further examining developmental changes throughout the lifespan. For the second, the initial elevation bias should be integrated more into the planning and evaluations of ambulatory assessment studies.

Since most items that were used in this project were originally developed for one-time assessment, an important goal for future research should be the development of well investigated questionnaires that can be used for daily measures, especially in self-report with children. These scales should prove to be valid, reliable, economic, minimally disruptive, not reactive, able to capture fluctuations in the daily experiences of participants, and ideally show accordance with objective measures [32]. Ambulatory assessment studies which can access such resources have the potential to capture important aspects of cognitive and behavioural functioning in humans.

Especially interesting for future research might be the results from our exploratory analyses. As we found that most of our hypotheses would have been confirmed, had we only considered the inattention factor of ADHD symptoms, this discrimination of the core symptoms in research should be pursued further.

## 5. Conclusions

In the current study we examined the association of sleep quality, daytime sleepiness, and ADHD symptoms in the daily life of German schoolchildren on an inter- and an intraindividual level. A significant negative intraindividual effect was found for daytime sleepiness on ADHD symptoms within participants, contrary to the hypotheses. Explorative analyses found significant effects of sleep and sleepiness on inattention on the inter- and the intraindividual level in the expected directions: Children who sleep better on average report less inattention; On days when children report better sleep, they indicate less inattention; And children who are sleepier on average during the day report more inattention. For hyperactivity/impulsivity we found an opposite effect to our expectations: in moments when children indicate to be sleepier during the day, they report less hyperactive/impulsive symptoms. We conclude that future research should preserve the advantages concerning ecological validity which the ambulatory assessment entails and possibly integrate it with the benefits that more objective measurements like actigraphy might add. Studies examining the precursors, correlations and effects of ADHD symptoms should split the construct in the two factors of attention and hyperactivity/impulsivity.

## Figures and Tables

**Figure 1 brainsci-12-00440-f001:**
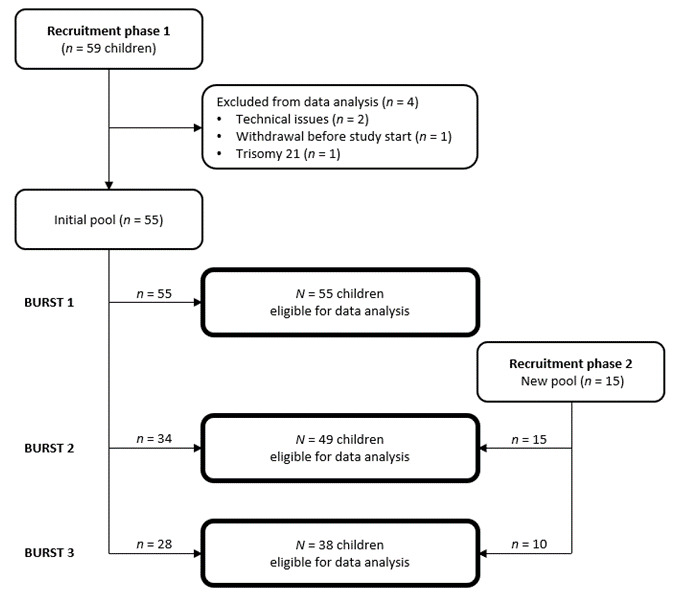
Recruitment process and retention of the participants.

**Figure 2 brainsci-12-00440-f002:**
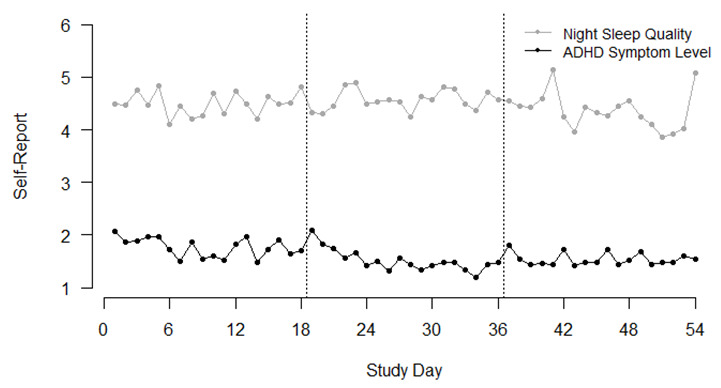
Time course of self-reported night sleep quality and ADHD symptom levels (in the afternoon) across all 54 study days; The dashed lines indicate breaks between each burst.

**Figure 3 brainsci-12-00440-f003:**
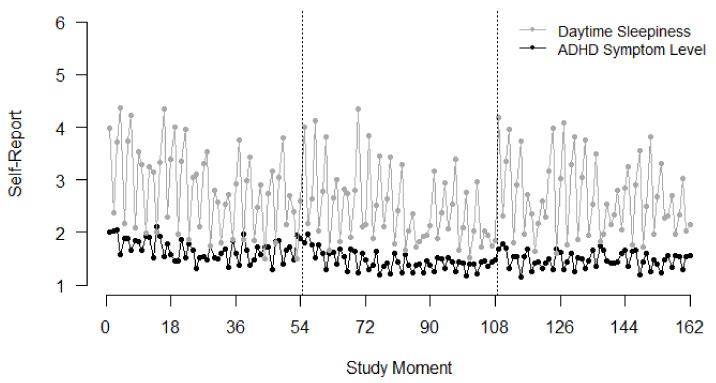
Time course of self-reported daytime sleepiness and ADHD symptom levels across all 162 study moments, with three moments per day; The dashed lines indicate breaks between each burst.

**Table 1 brainsci-12-00440-t001:** Descriptive statistics for children’s ADHD symptom levels, night sleep quality, and daytime sleepiness across all 54 study days.

	Between-Person			Within-Person			
	*M*	*SD*	Range	*M_ISD_*	*SD*	Range	ICC
ADHD (afternoon) ^1^	1.62	0.60	1.00–3.64	0.59	0.41	0.00–1.92	0.44
ADHD (overall) ^2^	1.52	0.50	1.00–2.91	0.56	0.37	0.00–1.79	0.43
Night sleep quality	4.58	0.82	2.23–6.00	1.11	0.57	0.00–2.37	0.43
Daytime sleepiness	2.68	0.98	1.00–4.42	1.48	0.55	0.00–2.28	0.34

*M* = mean, *SD* = standard deviation, *M_ISD_* = mean intra-individual standard deviation, ICC = intraclass correlation coefficient, theoretical range for all variables: 1–6, with higher values indicating higher ADHD symptom levels, better night sleep quality, and higher daytime sleepiness, respectively. ^1^ ADHD symptom level reports collected only at afternoon time points ^2^ ADHD symptom level reports collected at all three time points during a day.

**Table 2 brainsci-12-00440-t002:** Multilevel model to test the between- and within-person association between children’s night sleep quality and ADHD symptom levels the following day.

Fixed Effects		Estimate		*SE*	*p*
*Burst 1*					
Intercept: initial level	*γ* _00_	2.07		1.60	0.20
Time slope ^a^	*γ* _01_	−0.27	*	0.13	0.04
Night sleep quality, between-person differences	*γ* _02_	−0.16		0.09	0.07
Night sleep quality, within-person fluctuations	*γ* _03_	−0.02		0.02	0.40
Weekend effect	*γ* _04_	−0.02		0.06	0.75
*Change at Burst 2, compared to Burst 1*					
Change in level	*γ* _10_	−0.03		0.10	0.74
Change in time slope	*γ* _11_	−0.21		0.18	0.24
Change in effect of night sleep quality (between-person)	*γ* _12_	0.18	*	0.08	0.03
Change in effect of night sleep quality (within-person)	*γ* _13_	−0.002		0.04	0.95
Change in weekend effect	*γ* _14_	0.02		0.09	0.85
*Change at Burst 3, compared to Burst 1*					
Change in level	*γ* _20_	−0.25	*	0.11	0.02
Change in time slope	*γ* _21_	0.02		0.19	0.92
Change in effect of night sleep quality (between-person)	*γ* _22_	0.12		0.09	0.17
Change in effect of night sleep quality (within-person)	*γ* _23_	0.02		0.04	0.61
Change in weekend effect	*γ* _24_	−0.07		0.09	0.44
*Control variables*					
Gender	*γ* _30_	0.08		0.14	0.56
Age ^b^	*γ* _31_	−0.003		0.01	0.80
ADHD medication ^c^	*γ* _32_	0.29		0.23	0.22
*Random Effects & Covariances*		Estimate			*p* ^d^
*Level 2 (between-person)*					
Intercept: initial level	*SD(u* _0*i*_ *)*	0.62	***		<0.001
Time slope	*SD(u* _1*i*_ *)*	0.49			0.93
Sleep quality within-person fluctuations	*SD(u* _2*i*_ *)*	0.06			0.93
Intercept and time	*r(u* _0*i,*_ *u* _1*i*_ *)*	−0.51	**		0.003
Intercept and sleep quality fluctuations	*r(u* _0*i,*_ *u* _2*i*_ *)*	−0.78	*		0.03
Time and sleep quality fluctuations	*r(u* _1*i,*_ *u* _2*i*_ *)*	0.78			0.99
*Level 1 (within-person)*					
Residual	*SD*(*ε_it_*)	0.66			
Autocorrelation	*ρ*	0.34	***		<0.001

*N* = 70 children, *n* = 1450 considered observations, * *p* < 0.05, ** *p* < 0.01, *** *p* < 0.001. ^a^ Time is coded 0 = study day 1 within a measurement burst, 1 = study day 18 within a measurement burst, with equal intervals for the intervening study moments. ^b^ We were not able to collect data on one child’s age. To avoid it falling out from data analysis, we set its age to the sample mean age. ^c^ We were not able to collect data on one child’s medication status. To avoid it falling out from data analysis, we assumed it was not receiving ADHD medication. ^d^ The respective *p*-values for the random effect estimates were obtained by testing in pairs a model that includes the parameter in question against a model missing just this parameter via likelihood ratio tests.

**Table 3 brainsci-12-00440-t003:** Multilevel model to test the between- and within-person association between children’s daytime sleepiness and concurrent ADHD symptom levels.

Fixed Effects		Estimate		*SE*	*p*
*Burst 1*					
Intercept: initial level	*γ* _00_	1.65		1.13	0.14
Time slope ^a^	*γ* _01_	−0.23	**	0.08	0.004
Daytime sleepiness, between-person differences	*γ* _02_	0.16		0.10	0.11
Daytime sleepiness, within-person fluctuations	*γ* _03_	−0.04	*	0.02	0.01
Weekend effect	*γ* _04_	0.05		0.03	0.14
*Change at Burst 2, compared to Burst 1*					
Change in level	*γ* _10_	−0.16	**	0.06	0.006
Change in time slope	*γ* _11_	−0.06		0.10	0.57
Change in effect of daytime sleepiness (between-person)	*γ* _12_	−0.12	*	0.06	0.04
Change in effect of daytime sleepiness (within-person)	*γ* _13_	0.01		0.02	0.50
Change in weekend effect	*γ* _14_	−0.09		0.05	0.10
*Change at Burst 3, compared to Burst 1*					
Change in level	*γ* _20_	−0.28	***	0.06	<0.001
Change in time slope	*γ* _21_	0.06		0.10	0.57
Change in effect of daytime sleepiness (between-person)	*γ* _22_	−0.06		0.07	0.39
Change in effect of daytime sleepiness (within-person)	*γ* _23_	0.03		0.02	0.18
Change in weekend effect	*γ* _24_	−0.09		0.06	0.10
*Control variables*					
Gender	*γ* _30_	−0.002		0.10	0.98
Age ^b^	*γ* _31_	0.0003		0.009	0.97
ADHD medication ^c^	*γ* _32_	0.55	**	0.17	0.001
*Random Effects & Covariances*		Estimate			*p* ^d^
*Level 2 (between-person)*					
Intercept: initial level	*SD(u* _0*i*_ *)*	0.50	***		<0.001
Time slope	*SD(u* _1*i*_ *)*	0.40	***		<0.001
Sleepiness within-person fluctuations	*SD(u* _2*i*_ *)*	0.07	***		<0.001
Intercept and time	*r(u* _0*i,*_ *u* _1*i*_ *)*	−0.30			0.05
Intercept and sleepiness fluctuations	*r(u* _0*i,*_ *u* _2*i*_ *)*	−0.73	***		<0.001
Time and sleepiness fluctuations	*r(u* _1*i,*_ *u* _2*i*_ *)*	−0.04			0.54
*Level 1 (within-person)*					
Residual	*SD*(*ε_it_*)	0.64			
Autocorrelation	*ρ*	0.27	***		<0.001

*SE* = standard error, *N* = 70 children, *n* = 5559 considered observations, * *p* < 0.05, ** *p* < 0.01, *** *p* < 0.001. ^a^ Time is coded 0 = study day 1 within a measurement burst, 1 = study day 18 within a measurement burst, with equal intervals for the intervening study moments. ^b^ We were not able to collect data on one child’s age. To avoid it falling out from data analysis, we set its age to the sample mean age. ^c^ We were not able to collect data on one child’s medication status. To avoid it falling out from data analysis, we assume it was not receiving ADHD medication. ^d^ The respective *p*-values for the random effect estimates were obtained by testing in pairs a model that includes the parameter in question against a model missing just this parameter via likelihood ratio tests.

## Data Availability

The authors work together with the Leibniz Institute for Psychology Information (ZPID; http://www.zpid.de/index.php?lang=EN, access date 25 January 2022) to archive and document the data collected in the project leading to this study. The data gathered in this project will be made available for the scientific community two years after the final project report has been written.

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
