# Peer review of "The Association of Self-Reported ADHD Symptoms and Sleep in Daily Life of a General Population Sample of School Children: An Inter- and Intraindividual Perspective"

_brainsci, 2022, doi:10.3390/brainsci12040440_

Round 1

Reviewer 1 Report

The present paper examines the interplay of nightly sleep quality and sleepiness with symptoms of inattention and hyperactivity on the within- and between-person level in school children. Overall, the paper is well-written, the theoretical background and motivation are coherent, and the statistical analyses are sound and well-chosen (e.g., multilevel modeling with random effects). The study follows best-practice ideas (e.g., measurement burst design, pre-registration of hypotheses) and the questions that are raised and answered by this paper are relevant to the journal's readership. The findings contribute to a currently highly discussed research topic (i.e. within-person variability in self-regulation). My general evaluation of the paper is therefore positive. I only have a handful of issues, as this manuscript is already in a good shape. 

Introduction

  1. In its present form, the introduction focuses strongly on the dimensionality of and fluctuations in ADHD symptoms. Considering the general population sample, it would benefit from a broader perspective on variability in children’s self-regulation (e.g., Ludwig et al., 2016; Leonard et al., 2021; or other work) and less focus on medication (only received by 11.4% of the sample – so not the most crucial mechanism here).

Ludwig, K., Haindl, A., Laufs, R., & Rauch, W. A. (2016). Self-regulation in preschool children’s everyday life: Exploring day-to-day variability and the within- and between-person structure. Journal of Self-Regulation and Regulation, 2, 99–117. https://doi.org/10.11588/josar.2016.2.34357

Leonard, J., Lydon-Staley, D. M., Sharp, S. D. S., Liu, H. Z., Park, A., Bassett, D. S., et al. (2021). The toothbrushing task: A novel paradigm for studying daily fluctuations in young children’s persistence. Child Development. Advance online publication

Method

  1. Please indicate which multilevel reliability estimate you used.

Results

  1. I understand why you differentiate the bursts in your analyses and that your analyses plan was pre-registered. However, given the rather small sample size for each single burst and the moderate differences between bursts, wouldn’t it increase the statistical power to analyze all bursts together (54 days instead of 3*18 days)? At least as an explorative validation of the results?

Discussion

  1. Yes, the intervention should not play a major role for the present findings. However, I would repeat the analyses with a group dummy (implementation intentions no = 0/yes = 1) on the between-person level. It is plausible to assume that the major findings do not change, and I think this would benefit your manuscript (e.g., one sentence with a reference to the appendix).

Author Response

Dear Sir or Madam,

We thank you for your positive review of our manuscript and the helpful feedback you provided for possible improvement. In the following, we would like to respond to your comments and suggestions and explain our actions to implement them.

  1. In its present form, the introduction focuses strongly on the dimensionality of and fluctuations in ADHD symptoms. Considering the general population sample, it would benefit from a broader perspective on variability in children’s self-regulation (e.g., Ludwig et al., 2016; Leonard et al., 2021; or other work) and less focus on medication (only received by 11.4% of the sample – so not the most crucial mechanism here).

Response:

For the introduction, you proposed to set the focus more on the concept of self-regulation. We thank you very much for this valuable comment. We included the articles you suggested in our manuscript (page 2 of the manuscript). Furthermore, we modified the introduction to explain more clearly that we consider ADHD as a disorder of self-regulation. We are looking forward to hearing from you, whether you think these changes are sufficient to underline the direction of our analyses or whether in your opinion we might make some additional adjustments.

  1. Please indicate which multilevel reliability estimate you used.

Response:

To determine the reliability of the ADHD scores, we computed multilevel reliability estimates using generalizability theory analyses. We have now defined this more clearly in the method section of our manuscript (see pages 6-7).

  1. I understand why you differentiate the bursts in your analyses and that your analyses plan was pre-registered. However, given the rather small sample size for each single burst and the moderate differences between bursts, wouldn’t it increase the statistical power to analyze all bursts together (54 days instead of 3*18 days)? At least as an explorative validation of the results?

Response:

 In the described research project, the bursts were administered each half a year apart, which means that there were six months in between day 18 and day 19 of the study data. Therefore, we expected time dependent effects like data missing not at random to occur separately within the bursts. Our analyses of missing data confirm this view. We found that missing rates increased within each of the bursts by up to 6% compared to the previous day. The specific results are now depicted in the descriptive results of the manuscript (page 10). Thus, because these timeframes seem to play an important role in the structure of the missing data and this might in turn influence the relation of our variables in the model, we decided to stick to our current models with the bursts separated, although this might limit the power as you have pointed out in your review.

  1. Yes, the intervention should not play a major role for the present findings. However, I would repeat the analyses with a group dummy (implementation intentions no = 0/yes = 1) on the between-person level. It is plausible to assume that the major findings do not change, and I think this would benefit your manuscript (e.g., one sentence with a reference to the appendix).

Response:

Thank you for your last suggestion of including an additional model with the experimental condition as a control variable in the appendix. We followed your advice and as you assumed, it did not change the results for our hypothesized effects but strengthens our argumentation. The results of this additional analysis including the intervention as a group dummy are included as Appendix B in the revised manuscript.

In total, we find that your suggestions have added substantially to the quality of our paper and hope that you find the changes we made adequate.

We are looking forward to hearing your opinion on our revision!

Reviewer 2 Report

The study is a longitudinal study dealing with interindividual differences and intraindividual fluctuations over time between sleep quality, subjective sleepiness and ADHD symptoms in an unselected population of 5th grade children.

The study focuses on an interesting issue, however I have major concerns regarding the methodology adopted.

In particular my concerns regards:

1)The measures used:

a) the authors used modified scales and so not standardized measures (nor for the face-to-face assessment of children, nor for the remote administration).

b) Conners scales: which Conners did the authors use? On my knowledge self-report Conners scales are suitable for children older than those who participated in the study. For 10 years old children only parents and teachers reports are considered reliable.

c) Daytime sleepiness is measured through two items extracted form the Multidimentional Mood Questionnaire. There are known self-report pediatric scales for the measurement of daytime sleepiness (for example the PDSS by  Drake et al. 2003). The authors should motivate their choice.

In general the authors may furnish some data on the validity of the measures used (e.g. sensitivity of items selected; concurrent validity between face-to-face and remote assessment). In addition, it would be useful to know whether, at least in the starting phase of the study, any form of feedback was given to the children's answers (so that they could respond reliably) or whether any children were excluded from the sample because they seemed to answer with little awareness/reliability.

2) In the conclusion the authors briefly mention that after Burst 1 children were allocated to two different interventions aiming at improving self regulation. Why this intervention was mentioned only at the end of the study? The authors may explain better in the method section In what consisted these interventions and why they would be irrelevant for the Burst 2-3 results.

3) I find it difficult to follow the description of the analyses and results. In the method section, analyses are described to test hypotheses not mentioned in the introduction (e.g. symptomatology on weekends compared to school days). Also some results in the table remain uncommented in the text. The authors could simplify the description of the analyses and results by limiting themselves to the hypotheses announced in the introduction.

4) Finally, the authors have to mention the possibility that the data of Burst 2 and 3 may be compromised by a fatigue effect. The planned methodology may be too demanding for 10-year old children (who were interviewed three times a day for three periods of 18 days each).

Author Response

Dear Sir or Madam,

We thank you for your positive review of our manuscript and the helpful feedback you provided for possible improvement. In the following, we would like to respond to your comments and suggestions and explain our actions to implement them.

1)The measures used:

a) the authors used modified scales and so not standardized measures (nor for the face-to-face assessment of children, nor for the remote administration).

Response

We thank you for this important comment and totally agree with your view that standardized measures would have been preferable for this study. However, due to the ambulatory assessment design, we modified the scales to reduce participant burden by only selecting specific items. Unfortunately, there are no standardized measures available for ambulatory assessment studies with the types of research questions we had. We however conducted a pilot study to test the items first. Within the pilot study, seven children filled out all items of the Conners C3-AI (Lidzba et al., 2013) and all items of the Multidimensional Mood Questionnaire (Steyer et al., 1997) three times per day for one week. Only those items which showed considerable intraindividual fluctuations were included in the final study protocol. With this procedure, we unfortunately lowered the comparability of our results but at the same time lightened participant burden by not integrating items which don’t seem to be fit for repeated measurement.

b) Conners scales: which Conners did the authors use? On my knowledge self-report Conners scales are suitable for children older than those who participated in the study. For 10 years old children only parents and teachers reports are considered reliable.

Response:

The items used in the daily assessment were adapted from the children self-report scales of the Conners C3-AI (Lidzba et al., 2013) which, according to the authors, are appropriate for children between the age of 8 – 18 years. Reliabilities of the items were very high on the between-person level (RKF; .98-.99, see page 6) and good on the within-person level (RC; .61-.69, see page 7). In addition, the original, full children self-report scale of the Conners C3-AI was administered with every child at the beginning of each measurement burst, before daily assessment (baseline measure). To test for validity of the scales, we calculated the correlations between the baseline measures  and the daily measurements (individual average across all study days within the respective burst) and received small but significant correlations (r = 0.31 – 0.47, all p < 0.05, see page 7). In previous work, which is not published yet, we additionally calculated an overall correlation of the mean daily parent reports of the self-regulation of their child and the mean daily children reports. Self-regulation was defined by a combination of the Conners ADHD items and the self-control scale (SCS) (Bertrams & Dickhäuser, 2009). Correlation between the parent- and the children report of self-regulation was r = 0.39, p < .001. Since children at the age of 10-12 years spend much time at school or leisure activities without the parents, we would interpret this finding in a way that children and parents indicate the same construct of ADHD, but take different evidence into account. Taken all these analyses together, we would suggest that child reports in our study seem to reliably and validly measure ADHD symptoms. We however definitely agree with you that these self-report measures of children have to be interpreted cautiously and reliability has to be investigated more thoroughly and have added this argumentation to the limitations of our study (page 18).

c) Daytime sleepiness is measured through two items extracted form the Multidimentional Mood Questionnaire. There are known self-report pediatric scales for the measurement of daytime sleepiness (for example the PDSS by Drake et al. 2003). The authors should motivate their choice.

Response:

You point out that daytime sleepiness had better be measured with specific scales explicitly assessing this construct. We agree that for this current manuscript, other questionnaires might have been preferable. However, the focus of the study was on the self-regulation capacities of children in their daily lives and an extensive amount of possible antecedents, correlates, and consequences. To not exceed the  participant burden which was already high, we needed to restrict the number of questionnaires used. The Multidimensional Mood Questionnaire (Steyer et al., 1997) was selected since one of the general research questions in this project concerned affect of the children.  One of the subscales included in the questionnaire is activation, which explicitly asks for sleepiness in the participants. Although this seemed a good variable to work with in the analyses for this manuscript, we totally agree with your view that more specific standardized scales would enhance the evidence of the results. We would be happy to take this into consideration for our planning of future studies.

In general the authors may furnish some data on the validity of the measures used (e.g. sensitivity of items selected; concurrent validity between face-to-face and remote assessment).

Response:

Furthermore, as you suggested, we have now reported some more analyses on the validity of the measures we used. For the ADHD symptom scale, we were able to find small but significant correlations with the standardized measures of the ADHD index which the children filled out before each burst (r = 0.31 – 0.47, p< 0.05, see page 7). They completed these forms in the classroom with a researcher present. We concluded that the face-to-face assessment in the classroom would be considered as a measurement of the trait and the daily assessment via smartphone the state of ADHD symptoms. Our interpretation is that the small but significant correlation shows that the trait and the state measurements assess similar but not identical characteristics of ADHD. We thank you for this advice since our manuscript has profited from your suggestion.  

In addition, it would be useful to know whether, at least in the starting phase of the study, any form of feedback was given to the children's answers (so that they could respond reliably) or whether any children were excluded from the sample because they seemed to answer with little awareness/reliability.

Thank you for this question. We have now realized that the description of the study procedure was not sufficient and added a corresponding paragraph on pages 5-6 of our manuscript. Before each burst children received the smartphones at school which were used during the ambulatory assessment . All participants tested the questionnaires with a researcher present who could answer all occurring questions. It was emphasized that there were no right or wrong answers and that data would be analyzed pseudonymized. Therefore, we conclude, that children should have had all possibilities to answer honestly and truly depict their current experiences. Other than that, no feedback was given on the items, since they were considered to depict personal experience which the researches did not know about. For this reason also, no children were excluded from the sample only because of their answer tendencies. We were not able to tell whether for example especially low fluctuations in sleep quality were indicated because the children actually frequently slept well or whether they lacked the introspection in their sleep (see the discussion on pages 17-18 in our manuscript). We however definitely agree with you that this is an important limitation of our study and that future research should try to investigate this more thoroughly, for example by comparing objective and subjective measures of sleep and ADHD symptoms (an argument we highlighted in our discussion on pages 18-19).

2) In the conclusion the authors briefly mention that after Burst 1 children were allocated to two different interventions aiming at improving self regulation. Why this intervention was mentioned only at the end of the study? The authors may explain better in the method section In what consisted these interventions and why they would be irrelevant for the Burst 2-3 results.

Response:

We thank you for pointing out the need to further elaborate on the self-regulation intervention within our manuscript. We have now explained further in the Method section (page 6) that within the project an intervention to enhance self-regulation was conducted before Burst 2. Children were assigned to one of two groups, with the experimental group receiving the full intervention and the control group receiving a reduced intervention. Both groups showed slight improvement in their self-regulation with no significant difference between the groups. You can find an extensive description of the intervention in another article (Schwarz & Gawrilow, 2019). Since there was no difference between the groups, we did not expect the intervention group to influence the relationship between ADHD symptoms and sleep or daily sleepiness. . However, to strengthen our point, we reran the models with intervention as a control variable. As can be seen in Appendix B of the revised manuscript, none of the results concerning our hypotheses changes significantly due to the inclusion of the intervention. We hope that by doing so we adequately explained the interventions and why we do not consider them to be relevant for our research questions.

3) I find it difficult to follow the description of the analyses and results. In the method section, analyses are described to test hypotheses not mentioned in the introduction (e.g. symptomatology on weekends compared to school days). Also some results in the table remain uncommented in the text. The authors could simplify the description of the analyses and results by limiting themselves to the hypotheses announced in the introduction.

Response:

We agree with you that our models within the multilevel analyses seem very complex. To test whether we really need all time variables within the models (burst, weekend, day), we ran some additional analyses testing whether the structure of missing data was completely at random (see page 10 of the manuscript). Since we found that missing data was significantly higher on weekends (up to 68% in Burst 2) and that the probability of missing data increased significantly within each burst, we decided to leave the variables in the model. Within our revision, we have now tried to make it clearer, which variables have been included in our models because of the hypotheses we wanted to test (ADHD symptoms, night sleep and daytime sleepiness), which variables are time-variables that are included because of the specific data structure of the ambulatory assessment with measurement burst design (bursts and days), and which variables were control variables to account for missing data or group differences (weekend, age, gender, medication)(see pages 7-8). We hope that we have made the description of the analyses and the results clearer and more understandable by that.

4) Finally, the authors have to mention the possibility that the data of Burst 2 and 3 may be compromised by a fatigue effect. The planned methodology may be too demanding for 10-year old children (who were interviewed three times a day for three periods of 18 days each).

Response:

Thank you for pointing out this limitation, which we frequently find in ambulatory assessment designs. Our analyses show that there were indeed dropouts between the bursts (see figure 1 on page 5) and a linear trend of missing data within each burst (see page 10). This means that children missed more occasions with time (up to 6% more with every day of the burst) and that after each of the bursts some children decided that they did not want to participate any more. We tried to account for these effects by integrating time variables in our models (see pages 7-8). With respect to your suggestion, we have now described the effect of participant burden, dropouts and missing data more thoroughly in the discussion section of our manuscript (page 17).

In total, we find that your suggestions have added substantially to the quality of our paper and hope that you find the changes we made adequate.

We are looking forward to hearing your opinion on our revision!

Literature

Bertrams, A., & Dickhäuser, O. (2009). Messung dispositioneller Selbstkontroll-Kapazität. Eine deutsche Adaptation der Kurzform der Self-Control Scale (SCS-K-D). Diagnostica, 55(1), 2–10. https://doi.org/doi:10.1026/0012-1924.55.1.2

Lidzba, K., Christiansen, H., & Drechsler, R. (2013). Conners Skalen zu Aufmerksamkeit und Verhalten – 3 (Conners 3®). Deutschsprachige Adaption der Conners 3rd Edition® (Conners 3®) von C. Keith Conners. Verlag Hans Huber, Hogrefe AG.

Schwarz, U., & Gawrilow, C. (2019). Measuring and compensating for deficits of self-regulation in school children via ambulatory assessment. Psychology in Russia: State of the Art, 12(4), 8–22. https://doi.org/10.11621/pir.2019.0401

Steyer, R., Schwenkmezger, P., Notz, P., & Eid, M. (1997). Der Mehrdimensionale Befindlichkeitsfragebogen (MDBF). Hogrefe.